# Effect of Plate Configuration in the Primary Stability of Osteotomies and Biological Reconstructions of Femoral Defects: Finite-Element Study

**DOI:** 10.3390/bioengineering11050416

**Published:** 2024-04-24

**Authors:** M. A. Neto, M. F. Paulino, A. M. Amaro

**Affiliations:** University of Coimbra, CEMMPRE, ARISE, Department of Mechanical Engineering, 3030-788 Coimbra, Portugal; maria.paulino@dem.uc.pt (M.F.P.); ana.amaro@dem.uc.pt (A.M.A.)

**Keywords:** finite-element analysis, biological reconstruction, allograft, stability, osteosynthesis plates

## Abstract

Background/objective: Osteosynthesis is an alternative treatment for stabilizing femur-bone traumas. The initial stability of the fixation systems is one of the biomechanical parameters affecting implant failure and bone union, especially in surgeries of intercalary reconstructions after the removal of bone tumors. This study aimed to investigate the initial biomechanical effect of using one or two osteosynthesis plate configurations for femoral fixation and the effect of fastening the allograft to the osteosynthesis plate in the case of femoral allograft reconstructions. Methods: Three finite-element models of a femur with three different fixation conditions for a transverse osteotomy in the middle of the diaphysis, i.e., using one and two osteosynthesis plates and an intercalary allograft, were constructed. An eight-hole compression plate and a six-hole second plate were used to simulate osteosynthesis plates. The plate screws were tightened previously to the loading, and the tightening sequences simulate the bolt-tightening procedure in a surgical environment. The models were imported into the ADINA System for nonlinear analysis, using compression loads applied over the femur head. Results: Models with the dual fixation systems had the most outstanding compression stiffness. The femur head movement in the dual plate system was 24.8% smaller than in the single plate system. A statistical analysis of a region of interest (VOI) placed in the femur diaphysis showed that the biomechanical effect of using the dual plate system is smaller in the osteotomy region than at the femur head, e.g., a displacement average decrease of only 5% between the two systems, while the maximum value decreases by 26.8%. The allograft fixation to the second osteosynthesis plate leads to an improvement in the system stability. Conclusions: The results presented in this work show that including the bolt analysis in the femoral diaphysis osteotomy fixation will allow for capturing the nonlinear behavior of the osteotomy region more realistically. The stability of the intercalary reconstruction of the femoral diaphysis was higher when the allograft was fastened to the second osteosynthesis plate.

## 1. Introduction

With the rapid development of imaging and surgical technology, amputation has been routinely replaced by limb salvage, resulting in higher patient satisfaction following the surgical treatment of bone tumors [1]. In this procedure, bone allograft is widely used to avoid the length discrepancy in viable bone tissues or reconstruct shattered bones. Despite the limb salvage rate of 100% [1], the complication rate can also be 15%. Some of the reported postoperative complications include fatigue fracture of graft, delayed union, nonunion, infection, etc. [2,3,4].

The biomechanical reconstruction of the affected bone after resection is usually ensured using either metallic implants, biologic grafts with collagen membranes [5,6,7], or combined grafts. Endoprosthesis reconstruction is a good option for achieving initial stability postoperatively, but it has a long-term follow-up complication rate that can be considered prohibited for oncologic patients [8,9]. Moreover, when applied to young patients, they cannot adjust their function to the natural growth of limbs, which is inappropriate. Biologic reconstructions include the possibility of intercalary allografts, fibular grafts, bone transportation, and distraction osteogenesis [10]. The outcome of the biologic reconstruction depends on the patient’s osteogenic viability and the initial stability of the reconstruction. A prognostic with a favorable bone union has a lower risk of internal fixation breakage. During the healing phase, graft bone transfers loading and gradually replaces the stiffness lost by the plate. Nevertheless, an inadequate initial fixation may be a negative parameter contributing to the nonunion of graft and host bone, leading to internal fixation failure [10].

Internal fixation systems, as in the case of intramedullary nails and osteosynthesis plates, are the elected operative techniques to improve the initial stability after femoral intercalary defects. The elastic stable intramedullary nail is a treatment widely used in school-aged patients with femoral shaft fractures [11]. Nevertheless, plating has shown superior clinical outcomes in the case of length-unstable fractures [12]. Different plate configurations were gradually developed in clinical practice, including lateral bridging plate plus orthogonal adjuvant plates, lateral bridging plate plus medial adjuvant plate, and lateral bridging plate plus medial bridging plate [10,13,14]. To improve the initial stability of lower-extremities reconstructions, the two plates might be considered a better option for internal fixation [15].

This study compares the biomechanical outcome of patients with length-unstable femoral shaft fractures treated with one or two osteosynthesis plates and the effect of fastening the allograft to the osteosynthesis plate in the case of femoral allograft reconstructions. To the best of our knowledge, no finite-element study includes the analysis associated with the fixation procedure of the plate screws. Nevertheless, smooth surfaces simplified the thread screws, and the bolt diameter was defined as corresponding to the average diameter of the given thread.

## 2. Materials and Methods

### 2.1. Geometrical Models

Three-dimensional finite-element models of a femur with three different fixation conditions for a transverse osteotomy in the middle of the diaphysis, i.e., using one and two osteosynthesis plates and an allograft to reconstruct large-segment diaphysis defect, were constructed based on the model#3403 of Sawbones^®^. The CAD femur (model#3908, Sawbones, Vashon, WA, USA) was aligned at 11° in adduction and 9° in flexion, and the condyles were subtracted to the CAD geometry of a 120 × 93 × 114 m^3^ aluminum block, assuring that condyles were continuously connected to the aluminum block at approximately 94 mm of height [16].

The osteosynthesis plates were created using CAD tools (Solidworks^®^ 2014, Dassault Systèmes SOLIDWORKS Corp., Waltham, MA, USA) following the specifications defined by Kim et al. [17,18]. A compression plate with eight holes is used in all models, and a six-hole second plate is used to verify how the osteotomy stability is affected by the presence of a second plate. The screws used on both osteosynthesis plates are different and were created without including the thread fillets. Hence, screws with a diameter of 4.5 mm and a length of 32 mm were used on the eight-hole plate, while screws with a diameter of 3.5 mm and 25 mm of length allowed to assemble the six-hole plate and femur. The orientation of the screws relative to the plates is presented in Figure 1. It is worth noticing that the two screws of the eight-hole plate that are closest to the osteotomy have a different orientation relative to the femur axes, simulating the compression effect.

The femur model is equivalent to the left-hand-side human femur, and the eight-hole plate is assembled in the middle of the diaphysis on the greater trochanter side. The six-hole plate was rotated 80° to the right side of the femur in the direction of the head location; the two assemblies are presented in Figure 1a,c. In the allograft configuration, the femur was sectioned perpendicularly to its axe by two horizontal planes at 34 mm from each other. The allograft 3D body was placed in the middle of the femur diaphysis, as shown in Figure 1c.

Two different plates were created to apply the prescribed load uniformly over the femur head. The first plate was sketched in the horizontal plane, 40 × 40 mm^2^, tangent to the femur head, and vertically extruded to obtain a final thickness of 20 mm. Posteriorly, it was moved 10 mm in the axes femur direction to ensure that the femur head and plate had some parts overlap. The femur head was then trimmed from the plate using a Boolean operation. The second plate was also sketched in the horizontal plane, 80 × 80 mm^2^, tangent to the femur head, and vertically extruded to obtain a final thickness of 10 mm. The in-plane dimensions of the second plate are two times higher than the first plate to ensure that during loading, the contact between the femur head and plate is not lost due to higher femur head displacement. After the assembly of all components of the models, the solid bodies were exported in the format of single Parasolid binary files.

### 2.2. Numerical Models

Each one of the four models was imported into the ADINA^®^ System for linear and nonlinear finite-element analysis (ADINA AUI version 9.8, ADINA R&D Inc., Water-town, NY, USA). In all numerical models, the screws were tightened before loading. The screw-tightening sequences simulate the bolt-tightening procedure in a surgical environment. The preload was simulated using 3D-solid finite elements with the ‘bolt’ element option available in ADINA for simulation of different bolt handling procedures: bolt tensioning, i.e., the axial force in the bolt is specified; bolt shrinkage, i.e., the bolt shortening is specified. The material properties of the different components in the assembly are listed in Table 1

### 2.3. Bolt Analysis

The bolt type loadings were applied during the phase of bolt iterations, wherein the time is frozen, and the bolt parameters are iteratively adjusted. The bolt phase started using a simultaneous bolt shortening of 0.004 mm in all screws of the eight-hole plate, assuring that the osteosynthesis plate is the nearest possible to the bone interface. Posteriorly, the bolt tensile forces were applied incrementally, using increment values of 200 N until 1000 N was reached. The tightening sequences are illustrated in Figure 2, representing the sequence number of each screw on all four models. Preload values vary widely in the literature [18,21]; the value of 1000 N was chosen based on an average of some previous studies.

During the bolt analysis, it was essential to include the fixation of both bone femur parts, avoiding the rigid motion relative to the fixation system, because, in a surgical procedure, it is normal to fix the position of bones and plates through clamps. Hence, Figure 3 shows the surfaces where the prescribed displacement was set to zero. In the one-plate model, the zero displacement values were applied to an area of the proximal cortex and two surfaces of the eight-hole plate, as presented in Figure 3a. In the two-plate models, in addition to the conditions already mentioned, a zero displacement was also used on one of the surfaces of the six-hole plate, Figure 3b. Regarding the two-plate models using allografts, only the restrictions in the two plates were considered, as shown in Figure 3c.

### 2.4. Loading Analysis

The boundary conditions used during the bolt analysis were removed previously to the femur loading. Nevertheless, the rigid body motion of the femur was removed during the normal loading simulation, prescribing a zero displacement of all nodes at the bottom of the aluminum block that holds the femur condyles [16]. The loading magnitude was 300 N in the first loading scenario for the models with one and two osteosynthesis plates, 1000 N in the second loading scenario (SL), and the models with allografts. In both loading scenarios, the rigid displacements of the plate body in the x and z directions were removed. In contrast, the y displacement direction was used to promote the femur loading.

The glue mesh option was used to model the attachment of all pairs of components that remain perfectly bonded together, such as the inner surfaces of the cortical bone and the outer surfaces of the cancellous bone or the outer surface of the first load plate and the head femur. Nevertheless, the Lagrange multiplier technique was used to impose contact constraint conditions, assuring the possibility of relative motion between the surfaces of the following contact pairs: plate/screws; plate/bone; plate/allograft; bone/allograft.

Assignment of the mesh density to the solid bodies of each model was made by promoting equally spaced subdivisions of the bodies using a 2 mm element edge length. In some cases, the subdivision of specific faces was recalculated using an element length of 0.5 mm (500 μm) to promote a more refined mesh in areas requiring higher precision of the results. The areas where the mesh was refined were the screws, plates, and hole regions, with the rest being defined with a higher value. Discretization of the domains was assured by the Delaunay free-form meshing algorithm to generate eight-node hexahedral (brick) elements with mixed interpolation formulation (displacement and pressure-based), considering a constant pressure (1 degree of freedom). For elements with linear elastic material properties, additional displacement degrees of freedom were allowed by selecting the incompatible modes option.

## 3. Results

For a better understanding, the results are presented in two different sections: the first includes the displacements and stresses for the femur fixation without the presence of allograft, while the second presents those results for the case of using the allograft in the femur fixation.

### 3.1. Without Allograft Bone

The distribution of the magnitude of displacement is presented in Figure 4 for the first load scenario, i.e., a total load of 300 N, and for the second loading scenario, which uses only two osteosynthesis plates.

Maximum displacement occurs in the femur head in all models and loading scenarios. However, due to the higher level of stress in the osteotomy region, further analyses of the tissue displacements and stresses focused on a rectangular prism known as the volume of interest (VOI). This VOI is in the femoral diaphysis, with its length constrained by the dimensions of the first plate. The results of this analysis are presented in Table 2.

The results in Table 2 are displayed in a box plot graphic in Figure 5. The results of Figure 5b do not include the outlier values; otherwise, their values will blend the differences among smaller values.

### 3.2. With Allograft Bone

The results hereafter presented are related to the fixation of an allograft bone material with the mechanical properties of a healthy cortical bone. The allograft bone was placed in the middle of the diaphysis, and two different fixation scenarios were simulated: the allograft was not fixed to any of the two osteosynthesis plates, and alternatively, the allograft was fixed using two bolts inserted at the smaller osteosynthesis plate.

The numerical simulations of these two fixation scenarios, which include a total load of 1000 N applied over the second loading plate, stopped at the incremental 64-step due to excessive model deformation. Hence, the step corresponding to a 309 N load was the basis for model comparisons. The distribution of the displacement magnitude for this load is presented in Figure 6.

The displacement distribution over the allograft of both fixation systems is presented in Figure 7, and the contact tractions/compressions that are applied over both proximal and distal allograft surfaces are also presented in Figure 7. The results of the tissue von Mises stresses in the allograft bone for the case of a total load of 309 N are presented in Table 3.

The results of the tissue displacements and stresses on the cortical bone related to the VOI are presented in Table 4 for the case of a total load of 234 N; the box plots of those results are presented in Figure 8. The selection of a smaller load than the limit load, to which it was possible to obtain convergency on the model using the free allograft, was to avoid numerical local effects that can appear in distorted meshes or large displacement movements.

## 4. Discussion

Implant failure in femur reconstructions at midshaft using locked osteosynthesis plates is one of the most common problems after treating a bone defect resulting from a primary bone tumor or any other type of osteotomy. Presently, it is well accepted that locking plates for fixation of femoral shaft osteotomies provide higher stability than conventional osteosynthesis plates [22].

Hence, an eight-hole osteosynthesis-locked plate was selected as a primary fixation system, while the second osteosynthesis plate had only six holes and was shorter than the first plate. Moreover, the second plate was conventional, i.e., a non-located plate. The use of shorter plates is more appropriate for pediatric interventions [23]. Nevertheless, considerable controversy has arisen regarding the appropriate number of osteosynthesis plates required to ensure stability and avoid implant fracture [22]. Hence, the primary aim of this study was to investigate the biomechanical behavior differences produced using two osteosynthesis plates instead of only one for treating femoral shaft osteotomies [24]. Previous studies [22,25,26] have shown that double orthogonal locked plate constructions present higher stiffness than any other configuration. This work also adopted the orthogonal configuration, and the eight-hole plate was the only locked plate. In the research of Wisanuyotin et al. [26], the two plates also had two holes of difference, creating configurations with eighth and ten-hole plates. Hence, the plates used in the present work are smaller than those applied by Wisanuyotin et al. shorter plates indeed have relatively inferior fatigue properties than longer plates. Still, because the children have relatively smaller weight and height than adults, the eight-hole plate with the dimension of 135 × 16 × 5 m^3^ is similar to the Synthes 4.5 mm Narrow LCP locking plate, which is children-suitable [27]. The single-locking plate system is more widely used to threaten femoral shaft fractures in children than a dual-locking plate configuration [11,28]. Nevertheless, some authors suggest using dual-locking plate configurations to increase allograft stability and avoid fixation failure [22,26,29]. 

In this work, the finite-element analysis [16,30] was used to compare the biomechanical behavior differences between the two scenarios for midshaft femoral fixation using only one eight-hole locking plate placed at the lateral side of the femur. Alternatively, the eight-hole locking plate and a six-hole plate were placed at the lateral and anterior sides of the femur. The maximum values of the displacement for a total load of 300 N, which are plotted in Figure 4a,b, show that the femur head movement in the dual plate system is 24.8% smaller than in the single plate system. The magnitude of maximum values is ten times higher than those presented by Wisanuyotin et al. [26] for the axial compression case. Nevertheless, in their work, the variation of the displacement magnitude between the model with one lateral locking plate and the model with lateral and anterior locking plates was also 25.2%, i.e., this value is very close to 24.8%. Still, even though the magnitude of the displacement for the first loading scenario is ten times higher than the value presented by Wisanuyotin et al. [26] for the load of 300 N, when the second loading scenario is considered for a load of 1000 N, the difference between values became only of 4.7%. The variation of the main displacement differences can be related to a significant number of small differences, namely the dimension of the locking plates (4.5/5.0, broad Stainless-Steel Locking Compression Plate (LCP) System, DePuy Synthes, Raynham, MA, USA), the bolt fastening, the femur position, mechanical properties of the bones and due to the loading procedure. The effect of different loading conditions can be understood by comparing the maximum displacement of the dual system in both loading scenarios in this work; even though the total load of the second scenario is 3.3 times higher than in the first loading scenario, the maximum value of displacement in the VOI region was only 1.5 times higher than in the first loading condition. This behavior is related to the difference in the contact conditions between the loading plate and femur head applied on both loading scenarios.

The results presented in Table 2, related to the VOI region, show that the biomechanical effect of using the dual plate system is smaller in the osteotomy region than at the femur head. For instance, the displacement average decreases only 5% between the two systems, while the maximum value decreases 26.8%. Meanwhile, the average of von Mises stresses on the double system increases by 17.6% relative to the single system, and the maximum value increases even more, by 29%. Moreover, the information presented in Figure 5 also shows that the double system can be considered controversial for small loading intensities since the stability improvement is accompanied by tissue overloading. Nevertheless, the double system shows an interesting behavior for higher intensity loads; The average values grow 1.6 in the displacement magnitude and only 1.02 on the von Mises stress.

Biological reconstruction of large-segment defects in long bones is one of the first choices of orthopedic surgeons to repair bones and ensure the normal length of members. Nevertheless, these reconstructions are associated with high rates of complications and failures. Hence, evaluating the current problems of allograft reconstruction techniques and optimizing treatment strategies is imperative [14,31]. Better initial stability can promote bone healing and reduce the risk of nonunion or implant failure [10,15]. Hence, Paul and Abraham [13] combined a CFR-PEEK plate and nailing for intercalary resection with tibial allograft, concluding that larger comparative studies are needed to ensure that CarboFix CFR-PEEK implants may be safe and effective for intercalary resection. In a previous study related to the biological reconstruction with free fibular graft after resection, Li et al. concluded that two plates used as an internal fixation method are recommended for lower extremity reconstructions, especially in the femur. The autograft was always fixed using only the second plate in their procedure. Following this clinical assessment, results presented in Figure 6 show the effect of no allograft fixation on the femur stability: the displacement of the femur head grows 5.9%, whereas the displacement magnitude of the allograft grows 6.4%, as can be seen in Figure 7a,b. Moreover, the contact forces on the proximal and distal surfaces of the allograft are also presented in Figure 7c,d, showing the same pattern of distribution in both situations. Still, the maximum value is significantly higher in the free allograft case, about 161% higher. However, this value is in the edge contact segment that touches the first osteosynthesis plate and does not directly contribute to the contact improvement in the distal surface. The results of Figure 7e,f can confirm this idea on both distal and proximal contact surfaces. Figure 7e shows that the median and the average values of the contact tractions on the proximal free allograft surface are 35.6% and 6.4% smaller than in the case of fixed allograft. For the case of the distal contact surface, there is a high variation in the median value; it changed from 0 MPa to 3.25 MPa, while the average values changed only from 3.2 MPa in the free allograft to 3.72 MPa in the fixed case.

The drawback of creating holes on the allograft to allow its fixation is related to the stress level that the material is subjected to, as can be observed in the results presented in Table 3, wherein the statistical description of the von Mises stress values are presented for the allograft bone and all values are higher for the fixation case. The retrospective study of Goldin et al. [4] shows that patients with multiple plates of fixation had lower overall allograft survival, which can be related to the results presented in Table 3. Still, the results related to the box plots displacement and the von Mises stress intensities in the VOI, which are presented in Figure 8, show an improvement in the stability and of the stress level of the VOI when the fixation of allograft is introduced in the medical procedure. On average, numerical values showed an improvement of 11.14% in stability and about 14.40% in stress intensity.

The comparison between the results of the VOI region in the second load scenario for the double system without allograft and the double system with the allograft fixation, i.e., comparison of Figure 5 and Figure 8, shows that the mobility and the stress level of this region are significantly affected by the loss of material continuity: even with only 23.4% of the total load used in the double system without allograft, the grow of the maximum displacement was of 462.25% and in the maximum von Mises stress was of 60.48%. Moreover, this big mobility can also be related to the length discrepancy often appearing in plating procedures applied to younger patients [2]. Stable and congruent femoral diaphysis reconstructions are prerequisites for host-graft union [32]. However, fixations that are too rigid might increase the risk of late complications correlated with adverse bone remodeling [16,33]. Baleani et al. [32] developed a comparative study of three different host-graft junctions, using only one low-stiffness bone plate for the fixation system, except for the taper junction, where a low-stiffness intramedullary nail was used. An interesting result showed that taper junctions can be an alternative to improve the massive allograft stability without increasing the fixation system stiffness. Hence, future development of the present work can also include this taper junction to evaluate if it is a suitable system that can replace the double plate system.

This study’s limitations are related to simplifying the bolts used to perform the osteosynthesis plate fixation. To reduce the computational time and modeling difficulty, the simplified models reflect only the effects of the dominant factors on the behavior of bolted joints, holding the plates and bones together and transmitting the forces between them. The study did not account for the presence of soft tissue and only for bone; hence, the systems’ stability may differ. Even though surrounding tissue and muscle forces on the femur joints were not simulated, since this work is a comparative study between osteotomy fixation conditions that have always used the same model simplifications, it is expected that those simplifications lead to similar effects on all models.

## 5. Conclusions

The findings of this work show that in the femoral diaphysis osteotomy fixation using the dual fixation system, the femur head movement can be 24.8% smaller than when the single plate system was used. Nevertheless, the results also show that the double system can be considered controversial for small loading intensities since the stability improvement is also accompanied by tissue overloading. However, when comparing results, it is important to consider that different loading conditions between the femur head and the loading body might also be responsible for the nonlinear behavior of the load-displacement curve. Hence, when comparing the biomechanical behaviors of different models, it is also important to include their natural nonlinearity; nature rarely behaves linearly and less often follows the assumptions of linear mechanics. The results presented in this work also show that including the bolt analysis in the femoral diaphysis osteotomy fixation will allow the nonlinear behavior of the osteotomy region to be captured. The stability of the intercalary reconstruction of the femoral diaphysis was higher when the allograft was fastened to the second osteosynthesis plate, but creating holes on the allograft to allow its fixation is responsible for higher stress level on that region and, therefore, can diminish the cellular colonization of the allograft.

## Figures and Tables

**Figure 1 bioengineering-11-00416-f001:**
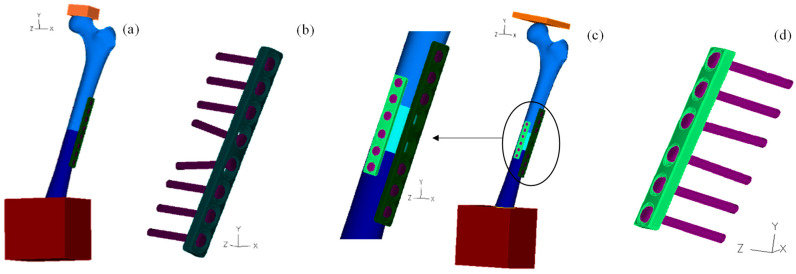
Femur fixed with osteosynthesis plates: (**a**) femur fixed with the eight-hole plate and loaded with the first loading plate; (**b**) configuration of the screws on the eight-hole plate; (**c**) femur with allograft positioned in the middle of the diaphysis and loaded with the second loading plate; (**d**) configuration of the screws on the six-hole plate.

**Figure 2 bioengineering-11-00416-f002:**
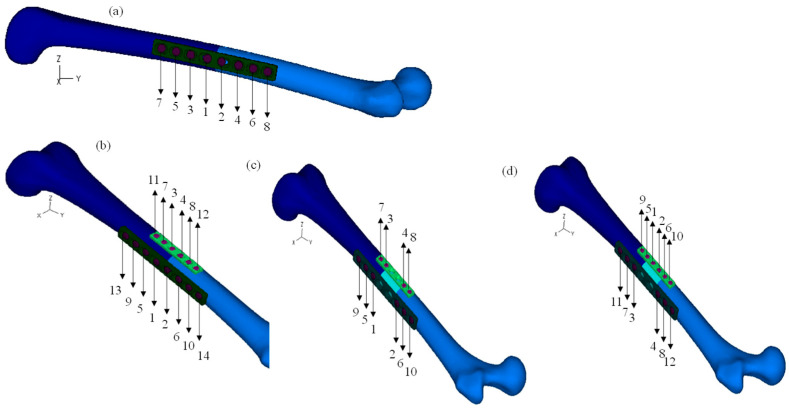
Tightening sequences: (**a**) one osteosynthesis plate; (**b**) two osteosynthesis plates; (**c**) first sequence for the two osteosynthesis plates and the allograft bone; (**d**) Second sequence for the two osteosynthesis plates and the allograft bone.

**Figure 3 bioengineering-11-00416-f003:**
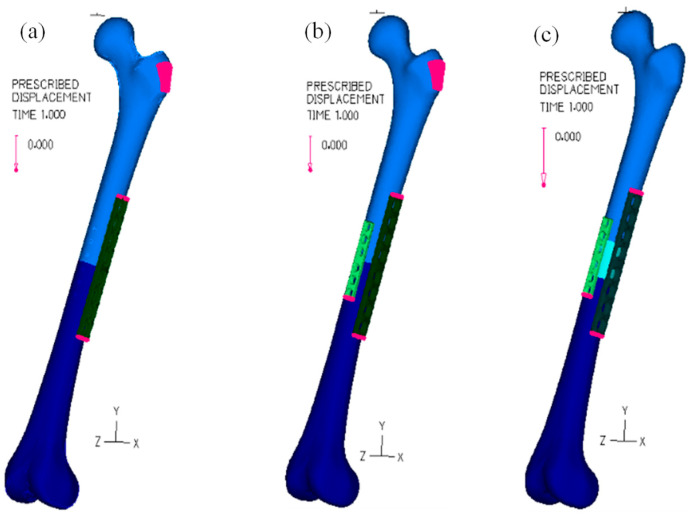
Prescribed displacement of different surfaces during the bolt analysis: (**a**) one osteosynthesis plate; (**b**) two osteosynthesis plates; (**c**) The osteosynthesis plates and the allograft bone.

**Figure 4 bioengineering-11-00416-f004:**
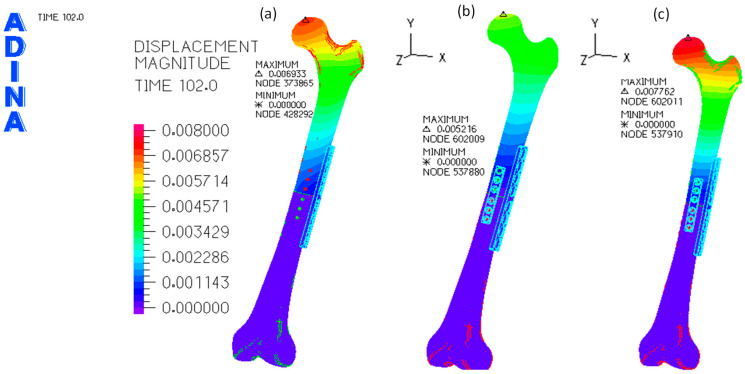
Distribution of the displacement magnitude (in meters) on the femur: (**a**) one osteosynthesis plate and the first load scenario; (**b**) two osteosynthesis plates and the first load scenario (300 N); (**c**) two osteosynthesis plates and the second load scenario (1000 N).

**Figure 5 bioengineering-11-00416-f005:**
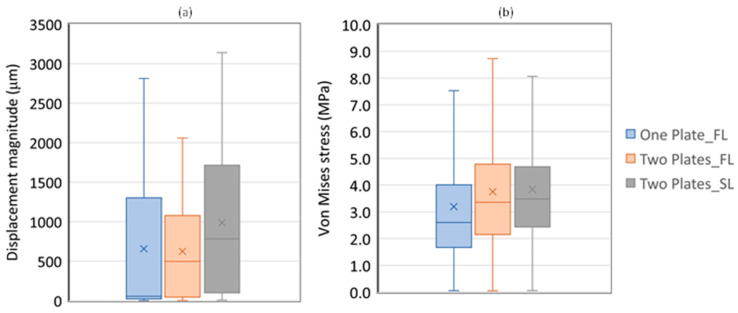
Box plot of results on the VOI: (**a**) displacement magnitude (in μm); (**b**) Von Mises stress (in MPa).

**Figure 6 bioengineering-11-00416-f006:**
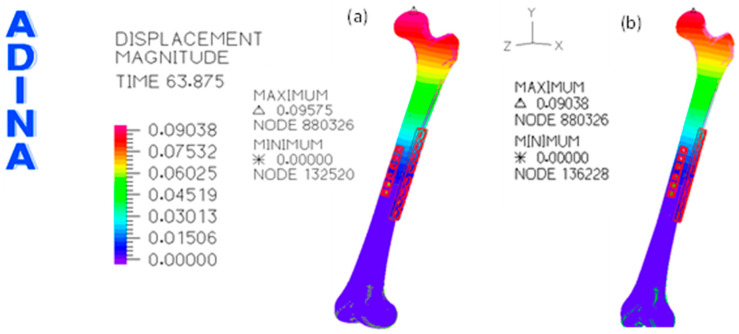
Distribution of the displacement magnitude (in meters) on the femur for the second load scenario: (**a**) for the free allograft case; (**b**) fixation of allograft with two bolts.

**Figure 7 bioengineering-11-00416-f007:**
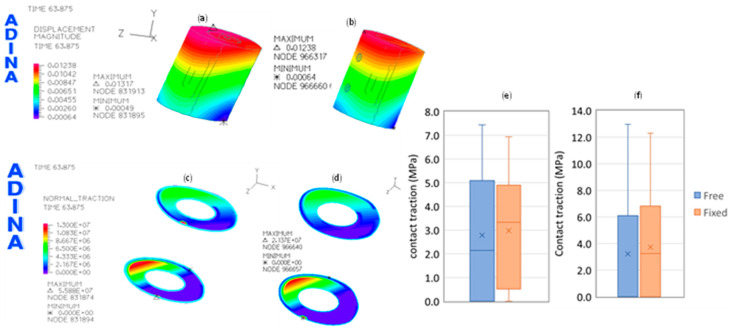
Displacement and contact forces result for the allograft bone when a load of 309 N is applied over the head femur: (**a**) displacement magnitude (in meters) on the free allograft; (**b**) displacement magnitude (in meters) on the fixed allograft; (**c**) contact traction on the proximal and distal allograft surfaces in the free allograft; (**d**) contact traction on the proximal and distal allograft surfaces in the fixed allograft; (**e**) box plots of the contact tractions in the proximal surface of free and fixed allograft conditions; (**f**) box plots of the contact tractions in the distal surface of free and fixed allograft conditions.

**Figure 8 bioengineering-11-00416-f008:**
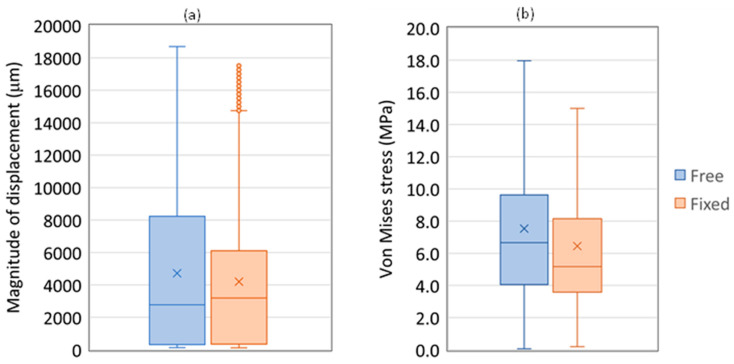
Box plots of the (**a**) displacement magnitude (in meters) and of the (**b**) von Mises stress on the femur for the second load scenario, using free and fixed allograft conditions.

**Table 1 bioengineering-11-00416-t001:** Mechanical properties of all materials.

Material	Density (kg/m^3^)	Young Modulus (GPa)	Coefficient of Poisson
Trabecular Bone, [19]	300	1.1	0.30
Cortical Bone, [19]	1800	15.0	0.30
screws and osteosynthesis plate (AISI 316L), [20]	8027	200.0	0.27
Aluminum loading plates	2700	69	0.33

**Table 2 bioengineering-11-00416-t002:** Statistical description of the distributions found for the VOI.

	Aver.	SD	Minimum	1st Quartile	Median	3rd Quartile	Maximum	Skewness	Kurtosis
Displacement (µm)									
One Plate	657.2	780.50	2.5	23.9	55.8	1301.1	2812.1	0.78	−0.80
Two Plates	623.7	573.9	1.8	44.9	498.1	1077.0	2058.3	0.59	−0.89
Two Plates_SL	987.7	897.3	5.9	98.7	781.4	1715.0	3139.2	0.59	−0.94
Von Mises (MPa)									
One Plate	3.2	2.8	0.1	1.7	2.6	4.0	83.8	6.3	90.0
Two Plates	3.8	2.7	0.1	2.2	3.4	4.8	106.5	6.1	108.6
Two Plates_SL	3.8	2.9	0.1	2.4	3.5	4.7	154.8	12.6	353.7

**Table 3 bioengineering-11-00416-t003:** Statistical description of the von Mises (MPa) distributions found for the allograft bone and a load of 309N.

	Aver.	SD	Minimum	1st Quartile	Median	3rd Quartile	Maximum	Skewness	Kurtosis
Free	3.7	2.3	0.2	1.6	3.5	5.3	32.5	1	4.4
Fixed	5.8	5.6	0.2	3.6	4.9	6.3	175.2	8.8	144.2
Variation (%)	58.6	145.5	−2.7	131.9	42.0	19.1	438.4	785.5	3196.7

**Table 4 bioengineering-11-00416-t004:** Statistical description of the distributions found for the VOI, including the allograft bone.

	Aver.	SD	Minimum	1st Quartile	Median	3rd Quartile	Maximum	Skewness	Kurtosis
Displacement (µm)									
Free	4723.	5102.	126.4	313.3	2771.0	8233.1	18,694.2	0.796	−0.647
Fixed	4197.	4210.	115.5	349.9	3194.1	6108.8	17,649.1	1.03	0.180
Von Mises (MPa)									
Free	7.5	5.6	0.1	4.1	6.7	9.6	258.4	5.6	132.3
Fixed	6.4	5.0	0.2	3.6	5.2	8.1	248.4	6.0	138.2

## Data Availability

http://hdl.handle.net/10316/87918.

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
