# Peer review of "Effect of Plate Configuration in the Primary Stability of Osteotomies and Biological Reconstructions of Femoral Defects: Finite-Element Study"

_bioengineering, 2024, doi:10.3390/bioengineering11050416_

Round 1

Reviewer 1 Report

Comments and Suggestions for Authors

This manuscript develops finite element models of a femur with different fixation for osteotomy, and investigates the effect of fixation configuration on the stability of osteosynthesis. The idea of the paper is valuable, presenting nice experiments and elaborated numerical work. However, there are several issues of this research.

Three critically important issues are as following:

1. For the clearer demonstration and better understanding of paragraph 4 in Section 2.1 (Page 3, Line 105-113), it’s advisable to add a figure illustrating how the loading plates are created.

2. I cannot find “the mesh density attributed to each body and the refinement areas” in Table 1 (Section 2.4, Page 5, Line 178-179).

3. The loading scenarios are confusing. Two magnitudes of loading (300N and 1000N) are adopted, as stated in Section 2.4 (Page 5, Line 163-165), however, two total loads (309N and 234N) are mentioned in Section 3.2. It’s advisable to revise the statement of loading conditions for each model, and add a table for clearer demonstration.

Other comments also need to be addressed:

4. The title of Figure 7b is incorrect (free allograft - fixed allograft); the title of Figure 8a uses wrong unit (meters - μm).

5. The description of fixation scenarios of allograft bone should be unified for consistency and disambiguation, for example, “free”, “unfixed”, “without fixation”, and “fixed”, “with fixation”.

In conclusion, the paper is worth publication, provided that the authors improve their manuscript in the light of the comments given.

Comments on the Quality of English Language

The authors are advised to thoroughly review and revise the entire paper for spelling and grammar. Maybe you could find a native English speaker or a professional language service to do a proof reading. Some of writing suggestions are listed below:

1. The symbol of decimal separator should be unified (“.” and “,”) (Section 2.1, Page 2, Line 88-90).

2. A “%” is missing in the statement “… the difference between values became only of 4.7” (Section 4, Page 10, Line 282-285).

3. The statement “following this clinical assessment the results presented in Figure 6 show the effect of no allograft fixation on the femur stability” (Section 4, Page 10, Line 320-321) should capitalize the first letter.

4. Please revise the statement “the displacement of the femur head it wil grow 5.9%, whereas the displacement magnitude of the allograft grows 6.4% as it can be seen in Figures 7a) and 7b).” (Section 4, Page 10, Line 321-323) for spelling and grammar.

5. The first letter of the statement “Nature rarely behaves linearly and less often follows the assumptions of linear mechanics.” (Section 5, Page 11, Line 378-379) should be lowercase.

Reviewer 2 Report

Comments and Suggestions for Authors

1. Although rigid body motion of the femur is mentioned to be removed during normal loading simulations, it is not explicitly stated how to deal with other possible boundary conditions (e.g. influence of surrounding soft tissues, limitation of joint motion, etc.)

2. The Lagrange multiplier method was used to impose contact constraints, but the authors did not indicate specific contact pairs and corresponding contact models in detail , and lacked a detailed description of the contact behavior.

3. For elements with linear elastic material properties, although the authors selected the incompatible mode option, the specific material parameters and how these were calibrated to match the experimental data were not indicated.

4. In clinical context, the stress after femoral internal fixation is generally concentrated at the fracture break. From Figure 4,6, the stress is concentrated in the femoral head. So in the finite element analysis, is it necessary to add more conditions, such as adjacent joints?

5. The following references on regenerative medicine might be helpful to provide more information and benefit the future readers: e.g., Engineered Regeneration 3 (2023) 217–231./ Engineered Regeneration 2022, 3 (1), 80-91/ Smart Medicine 2022, 1 (1), e20220006./ Smart Medicine 2022, 1 (1), e20220012./ Biomedical Technology 2023, 2, 31-48. / Biomedical Technology 2023, 2, 1-14.

6. Allograft bone is often used as cancellous bone for implantation of fracture breaks, does this have an impact on the setting of boundary conditions and experimental results.

Comments on the Quality of English Language

None

Reviewer 3 Report

Comments and Suggestions for Authors

Recommendation: Major revisions needed as noted.

Comments:

This study aimed to investigate the initial biome-chanical effect of using one or two osteosynthesis plate configurations for femoral fixation and the effect of fastening the allograft to the osteosynthesis plate in the case of femoral allograft reconstructions. This research work is very interesting, and the reviewer has some questions and suggestions that the author should take on board.   1. The abstract section is irregular. The research system, main methods, important findings and core conclusions of this paper should be summarized briefly and succinctly. 2. The first paragraph of the introduction is not discussed the progress of this study and the characteristics of the job. 3. The “discussion” section should be discussed in conjunction with results. 4. Each chapter should discuss the internal relationship between the other, which can enhance the logic of the paper.

5. The paper may be helpful for enriching the content of the manuscript. (DOI: 10.1016/j.compositesb.2022.109864)

6. There are spelling errors and grammar problems in this paper. I suggest author to seek help from experts to modify English.

Comments on the Quality of English Language

There are spelling errors and grammar problems in this paper. I suggest author to seek help from experts to modify English.

Round 2

Reviewer 3 Report

Comments and Suggestions for Authors

Continue to revise abstracts and English.

Comments on the Quality of English Language

Continue to revise English.

Author Response

We appreciate the time and effort you dedicated to providing feedback on our manuscript and are grateful for our paper's precise, constructive, and valuable review. The changes made are highlighted in blue in the revised manuscript. 

best regards